# Wasserstein Learning of Deep Generative Point Process Models

**Shuai Xiao**[*][†]**, Mehrdad Farajtabar**[*][◇] **Xiaojing Ye**[‡]**, Junchi Yan**[§]**, Le Song**[◇][¶]**, Hongyuan Zha**[◇]

[†] Shanghai Jiao Tong University
[◇]College of Computing, Georgia Institute of Technology
[‡] School of Mathematics, Georgia State University
[§] IBM Research – China
[¶]Ant Financial
benjaminforever@sjtu.edu.cn, mehrdad@gatech.edu
xye@gsu.edu, yanjc@cn.ibm.com
{lsong,zha}@cc.gatech.edu

## Abstract

Point processes are becoming very popular in modeling asynchronous sequential data due to their sound mathematical foundation and strength in modeling a variety of real-world phenomena. Currently, they are often characterized via intensity function which limits model's expressiveness due to unrealistic assumptions on its parametric form used in practice. Furthermore, they are learned via maximum likelihood approach which is prone to failure in multi-modal distributions of sequences. In this paper, we propose an intensity-free approach for point processes modeling that transforms nuisance processes to a target one. Furthermore, we train the model using a likelihood-free leveraging Wasserstein distance between point processes. Experiments on various synthetic and real-world data substantiate the superiority of the proposed point process model over conventional ones.

## 1 Introduction

Event sequences are ubiquitous in areas such as e-commerce, social networks, and health informatics. For example, events in e-commerce are the times a customer purchases a product from an online vendor such as Amazon. In social networks, event sequences are the times a user signs on or generates posts, clicks, and likes. In health informatics, events can be the times when a patient exhibits symptoms or receives treatments. Bidding and asking orders also comprise events in the stock market. In all of these applications, understanding and predicting user behaviors exhibited by the event dynamics are of great practical, economic, and societal interest.

Temporal point processes [1] is an effective mathematical tool for modeling events data. It has been applied to sequences arising from social networks [2, 3, 4], electronic health records [5], e-commerce [6], and finance [7]. A temporal point process is a random process whose realization consists of a list of discrete events localized in (continuous) time. The point process representation of sequence data is fundamentally different from the discrete time representation typically used in time series analysis. It directly models the time period between events as random variables, and allows temporal events to be modeled accurately, without requiring the choice of a time window to aggregate events, which may cause discretization errors. Moreover, it has a remarkably extensive theoretical foundation [8].

---

[*]Authors contributed equally. Work completed at Georgia Tech.

However, conventional point process models often make strong unrealistic assumptions about the generative processes of the event sequences. In fact, a point process is characterized by its *conditional intensity function* – a stochastic model for the time of the next event given all the times of previous events. The functional form of the intensity is often designed to capture the phenomena of interests [9]. Some examples are homogeneous and non-homogeneous Poisson processes [10], self-exciting point processes [11], self-correcting point process models [12], and survival processes [8]. Unfortunately, they make various parametric assumptions about the latent dynamics governing the generation of the observed point patterns. As a consequence, model misspecification can cause significantly degraded performance using point process models, which is also shown by our experimental results later.

To address the aforementioned problem, the authors in [13, 14, 15] propose to learn a general representation of the underlying dynamics from the event history without assuming a fixed parametric form in advance. The intensity function of the temporal point process is viewed as a nonlinear function of the history of the process and is parameterized using a recurrent neural network. Attenional mechanism is explored to discover the underlying structure [16]. Apparently this line of work still relies on explicit modeling of the intensity function. However, in many tasks such as data generation or event prediction, knowledge of the whole intensity function is unnecessary. On the other hand, sampling sequences from intensity-based models is usually performed via a thinning algorithm [17], which is computationally expensive; many sample events might be rejected because of the rejection step, especially when the intensity exhibits high variation. More importantly, most of the methods based on intensity function are trained by maximizing log likelihood or a lower bound on it. They are asymptotically equivalent to minimizing the Kullback-Leibler (KL) divergence between the data and model distributions, which suffers serious issues such as mode dropping [18, 19]. Alternatively, Generative Adversarial Networks (GAN) [20] have proven to be a promising alternative to traditional maximum likelihood approaches [21, 22].

In this paper, for the first time, we bypass the intensity-based modeling and likelihood-based estimation of temporal point processes and propose a neural network-based model with a generative adversarial learning scheme for point processes. In GANs, two models are used to solve a minimax game: a generator which samples synthetic data from the model, and a discriminator which classifies the data as real or synthetic. Theoretically speaking, these models are capable of modeling an arbitrarily complex probability distribution, including distributions over discrete events. They achieve state-of-the-art results on a variety of generative tasks such as image generation, image super-resolution, 3D object generation, and video prediction [23, 24].

The original GAN in [20] minimizes the Jensen-Shannon (JS) and is regarded as highly unstable and prone to miss modes. Recently, Wasserstein GAN (WGAN) [25] is proposed to use the Earth Moving distance (EM) as an objective for training GANs. Furthermore it is shown that the EM objective, as a metric between probability distributions [26] has many advantages as the loss function correlates with the quality of the generated samples and reduces mode dropping [27]. Moreover, it leverages the geometry of the space of event sequences in terms of their distance, which is not the case for an MLE-based approach. In this paper we extend the notion of WGAN for temporal point processes and adopt a Recurrent Neural Network (RNN) for training. Importantly, we are able to demonstrate that Wasserstein distance training of RNN point process models outperforms the same architecture trained using MLE.

In a nutshell, the contributions of the paper are: i) We propose the first intensity-free generative model for point processes and introduce the first (to our best knowledge) likelihood-free corresponding learning methods; ii) We extend WGAN for point processes with Recurrent Neural Network architecture for sequence generation learning; iii) In contrast to the usual subjective measures of evaluating GANs we use a statistical and a quantitative measure to compare the performance of the model to the conventional ones. iv) Extensive experiments involving various types of point processes on both synthetic and real datasets show the promising performance of our approach.

## 2  Proposed Framework

In this section, we define Point Processes in a way that is suitable to be combined with the WGANs.

### 2.1  Point Processes

Let $S$ be a compact space equipped with a Borel $\sigma$-algebra $\mathcal{B}$. Take $\Xi$ as the set of counting measures on $S$ with $\mathcal{C}$ as the smallest $\sigma$-algebra on it. Let $(\Omega, \mathcal{F}, \mathbb{P})$ be a probability space. A point process on $S$ is a measurable map $\xi : \Omega \to \Xi$ from the probability space $(\Omega, \mathcal{F}, \mathbb{P})$ to the measurable space $(\Xi, \mathcal{C})$. Figure 1-a illustrates this mapping.

Every *realization* of a point process $\xi$ can be written as $\xi = \sum_{i=1}^{n} \delta_{X_i}$ where $\delta$ is the Dirac measure, $n$ is an integer-valued random variable and $X_i$'s are random elements of $S$ or *events*. A point process can be equivalently represented by a *counting process*: $N(B) := \int_B \xi(x)dx$, which basically is the number of events in each Borel subset $B \in \mathcal{B}$ of $S$. The *mean measure* $M$ of a point process $\xi$ is a measure on $S$ that assigns to every $B \in \mathcal{B}$ the expected number of events of $\xi$ in $B$, *i.e.*, $M(B) := \mathbb{E}[N(B)]$ for all $B \in \mathcal{B}$.

For *inhomogeneous Poisson process*, $M(B) = \int_B \lambda(x)dx$, where the intensity function $\lambda(x)$ yields a positive measurable function on $S$. Intuitively speaking, $\lambda(x)dx$ is the expected number of events in the infinitesimal $dx$. For the most common type of point process, a *Homogeneous Poisson process*, $\lambda(x) = \lambda$ and $M(B) = \lambda|B|$, where $|\cdot|$ is the Lebesgue measure on $(S, \mathcal{B})$. More generally, in *Cox point processes*, $\lambda(x)$ can be a random density possibly depending on the history of the process. For any point process, given $\lambda(\cdot)$, $N(B) \sim \text{Poisson}(\int_B \lambda(x)dx)$. In addition, if $B_1, \ldots, B_k \in \mathcal{B}$ are disjoint, then $N(B_1), \ldots, N(B_k)$ are independent conditioning on $\lambda(\cdot)$.

For the ease of exposition, we will present the framework for the case where the events are happening in the real half-line of time. But the framework is easily extensible to the general space.

## 2.2 Temporal Point Processes

A particularly interesting case of point processes is given when $S$ is the time interval $[0, T]$, which we will call a *temporal point process*. Here, a realization is simply a set of time points: $\xi = \sum_{i=1}^{n} \delta_{t_i}$. With a slight notation abuse we will write $\xi = \{t_1, \ldots, t_n\}$ where each $t_i$ is a random time before $T$. Using a conditional intensity (rate) function is the usual way to characterize point processes.

For Inhomogeneous Poisson process (**IP**), the intensity $\lambda(t)$ is a fixed non-negative function supported in $[0, T]$. For example, it can be a multi-modal function comprised of $k$ Gaussian kernels: $\lambda(t) = \sum_{i=1}^{k} \alpha_i (2\pi\sigma_i^2)^{-1/2} \exp\left(-(t - c_i)^2/\sigma_i^2\right)$, for $t \in [0, T)$, where $c_i$ and $\sigma_i$ are fixed center and standard deviation, respectively, and $\alpha_i$ is the weight (or importance) for kernel $i$.

A self-exciting (Hawkes) process (**SE**) is a cox process where the intensity is determined by previous (random) events in a special parametric form: $\lambda(t) = \mu + \beta \sum_{t_i < t} g(t - t_i)$, where $g$ is a nonnegative kernel function, *e.g.*, $g(t) = \exp(-\omega t)$ for some $\omega > 0$. This process has an implication that the occurrence of an event will increase the probability of near future events and its influence will (usually) decrease over time, as captured by (the usually) decaying fixed kernel $g$. $\mu$ is the exogenous rate of firing events and $\alpha$ is the coefficient for the endogenous rate.

In contrast, in self-correcting processes (**SC**), an event will decrease the probability of an event: $\lambda(t) = \exp(\eta t - \sum_{t_i < t} \gamma)$. The exp ensures that the intensity is positive, while $\eta$ and $\gamma$ are exogenous and endogenous rates.

We can utilize more flexible ways to model the intensity, *e.g.*, by a Recurrent Neural Network (**RNN**): $\lambda(t) = g_w(t, h_{t_i})$, where $h_{t_i}$ is the feedback loop capturing the influence of previous events (last updated at the latest event) and is updated by $h_{t_i} = h_v(t_i, h_{t_{i-1}})$. Here $w$, $v$ are network weights.

## 2.3 Wasserstein-Distance for Temporal Point Processes

Given samples from a point process, one way to estimate the process is to find a model $(\Omega_g, \mathcal{F}_g, \mathbb{P}_g) \to (\Xi, \mathcal{C})$ that is *close enough* to the real data $(\Omega_r, \mathcal{F}_r, \mathbb{P}_r) \to (\Xi, \mathcal{C})$. As mentioned in the introduction, Wasserstein distance [25] is our choice as the proximity measure. The Wasserstein distance between distribution of two point processes is:

$$W(\mathbb{P}_r, \mathbb{P}_g) = \inf_{\psi \in \Psi(\mathbb{P}_r, \mathbb{P}_g)} \mathbb{E}_{(\xi, \rho) \sim \psi}[\|\xi - \rho\|_\star], \tag{1}$$

where $\Psi(\mathbb{P}_r, \mathbb{P}_g)$ denotes the set of all joint distributions $\psi(\xi, \rho)$ whose marginals are $\mathbb{P}_r$ and $\mathbb{P}_g$.

The distance between two sequences $\|\xi - \rho\|_\star$, is tricky and need further attention. Take $\xi = \{x_1, x_2, \ldots, x_n\}$ and $\rho = \{y_1, \ldots, y_m\}$, where for simplicity we first consider the case $m = n$. The two sequences can be thought as discrete distributions $\mu^\xi = \sum_{i=1}^{n} \frac{1}{n} \delta_{x_i}$ and $\mu^\rho = \sum_{i=1}^{n} \frac{1}{n} \delta_{y_i}$. Then, the distance between these two is an optimal transport problem $\text{argmin}_{\pi \in \Sigma} \langle \pi, C \rangle$, where $\Sigma$ is the set of doubly stochastic matrices (rows and columns sum up to one), $\langle \cdot, \cdot \rangle$ is the Frobenius dot product, and $C$ is the cost matrix. $C_{ij}$ captures the energy needed to move a probability mass from $x_i$ to $y_j$. We take $C_{ij} = \|x_i - y_j\|_\circ$ where $\|\cdot\|_\circ$ is the norm in $S$. It can be seen that the optimal solution is attained at extreme points and, by Birkhoff's theorem, the extreme points of the set of doubly stochastic matrices is a permutation [28]. In other words, the mass is transfered from a unique source

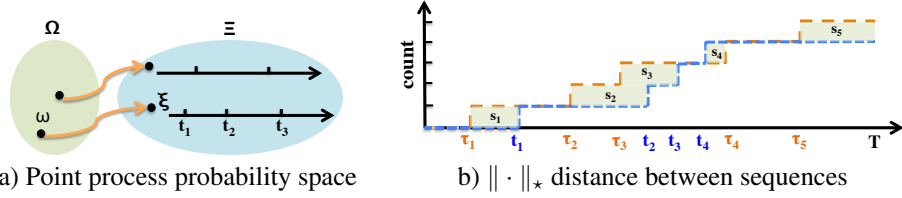

a) Point process probability space          b) $\| \cdot \|_\star$ distance between sequences

Figure 1: a) The outcome of the random experiment $\omega$ is mapped to a point in space of count measures $\xi$; b) Distance between two sequences $\xi = \{t_1, t_2, \ldots\}$ and $\rho = \{\tau_1, \tau_2, \ldots\}$

event to a unique target event. Therefore, we have: $\|\xi - \rho\|_\star = \min_\sigma \sum_{i=1}^n \|x_i - y_{\sigma(i)}\|_\circ$, where the minimum is taken among all $n!$ permutations of $1 \ldots n$. For the case $m \neq n$, without loss of generality we assume $n \leq m$ and define the distance as follows:

$$\|\xi - \rho\|_\star = \min_\sigma \sum_{i=1}^n \|x_i - y_{\sigma(i)}\|_\circ + \sum_{i=n+1}^m \|s - y_{\sigma(i)}\|, \tag{2}$$

where $s$ is a fixed limiting point in border of the compact space $S$ and the minimum is over all permutations of $1 \ldots m$. The second term penalizes unmatched points in a very special way which will be clarified later. Appendix B proves that it is indeed a valid distance measure.

Interestingly, in the case of temporal point process in $[0, T)$ the distance between $\xi = \{t_1, \ldots, t_n\}$ and $\rho = \{\tau_1, \ldots, \tau_m\}$ is reduced to

$$\|\xi - \rho\|_\star = \sum_{i=1}^n |t_i - \tau_i| + (m - n) \times T - \sum_{i=n+1}^m \tau_i, \tag{3}$$

where the time points are ordered increasingly, $s = T$ is chosen as the anchor point, and $|\cdot|$ is the Lebesgue measure in the real line. A proof is given in Appendix C. This choice of distance is significant in two senses. First, it is computationally efficient and no excessive computation is involved. Secondly, in terms of point processes, it is interpreted as the volume by which the two counting measures differ. Figure 1-b demonstrates this intuition and justifies our choice of metric in $\Xi$ and Appendix D contains the proof. The distance used in our current work is the simplest yet effective distance that exhibits high interpretability and efficient computability. More robust distance like local alignment distance and dynamic time warping [29] should be more robust and are great venues for future work.

Equation (1) is computationally highly intractable and its dual form is usually utilized [25]:

$$W(\mathbb{P}_r, \mathbb{P}_g) = \sup_{\|f\|_L \leq 1} \mathbb{E}_{\xi \sim \mathbb{P}_r}[f(\xi)] - \mathbb{E}_{\rho \sim \mathbb{P}_g}[f(\rho)], \tag{4}$$

where the supremum is taken over all Lipschitz functions $f : \Xi \to \mathbb{R}$, *i.e.*, functions that assign a value to a sequence of events (points) and satisfy $|f(\xi) - f(\rho)| \leq \|\xi - \rho\|_\star$ for all $\xi$ and $\rho$.

However, solving the dual form is still highly nontrivial. Enumerating all Lipschitz functions over point process realizations is impossible. Instead, we choose a parametric family of functions to approximate the search space $f_w$ and consider solving the problem

$$\max_{w \in \mathcal{W}, \|f_w\|_L \leq 1} \mathbb{E}_{\xi \sim \mathbb{P}_r}[f_w(\xi)] - \mathbb{E}_{\rho \sim \mathbb{P}_g}[f_w(\rho)] \tag{5}$$

where $w \in \mathcal{W}$ is the parameter. The more flexible $f_w$, the more accurate will be the approximation.

It is notable that W-distance leverages the geometry of the space of event sequences in terms of their distance, which is not the case for MLE-based approach. It in turn requires functions of event sequences $f(x_1, x_2, \ldots)$, rather than functions of the time stamps $f(x_i)$. Furthermore, Stein's method to approximate Poisson processes [30, 31] is also relevant as they are defining distances between a Poisson process and an arbitrary point process.

## 2.4   WGAN for Temporal Point Processes
Equipped with a way to approximately compute the Wasserstein distance, we will look for a model $\mathbb{P}_r$ that is close to the distribution of real sequences. Again, we choose a sufficiently flexible parametric family of models, $g_\theta$ parameterized by $\theta$. Inspired by GAN [20], this generator takes a noise and turns it into a sample to mimic the real samples. In conventional GAN or WGAN, Gaussian or uniform distribution is chosen. In point processes, a homogeneous Poisson process plays the role of a

non-informative and uniform-like distribution: the probability of events in every region is independent of the rest and is proportional to its volume. Define the noise process as $(\Omega_z, \mathcal{F}_z, \mathbb{P}_z) \to (\Xi, \mathcal{C})$, then $\zeta \sim \mathbb{P}_z$ is a sample from a Poisson process on $S = [0, T]$ with constant rate $\lambda_z > 0$. Therefore, $g_\theta : \Xi \to \Xi$ is a transformation in the space of counting measures. Note that $\lambda_z$ is part of the prior knowledge and belief about the problem domain. Therefore, the objective of learning the generative model can be written as $\min W(\mathbb{P}_r, \mathbb{P}_g)$ or equivalently:

$$\min_{\theta} \max_{w \in \mathcal{W}, \|f_w\|_L \leq 1} \mathbb{E}_{\xi \sim \mathbb{P}_r}[f_w(\xi)] - \mathbb{E}_{\zeta \sim \mathbb{P}_z}[f_w(g_\theta(\zeta))] \tag{6}$$

In GAN terminology $f_w$ is called the *discriminator* and $g_\theta$ is known as the *generator* model. We estimate the generative model by enforcing that the sample sequences from the model have the same distribution as training sequences. Given $L$ samples sequences from real data $\mathcal{D}_r = \{\xi_1, \ldots, \xi_L\}$ and from the noise $\mathcal{D}_z = \{\zeta_1, \ldots, \zeta_L\}$ the two expectations are estimated empirically: $\mathbb{E}_{\xi \sim \mathbb{P}_r}[f_w(\xi)] = \frac{1}{L} \sum_{l=1}^{L} f_w(\xi_l)$ and $\mathbb{E}_{\zeta \sim \mathbb{P}_z}[f_w(g_\theta(\zeta))] = \frac{1}{L} \sum_{l=1}^{L} f_w(g_\theta(\zeta_l))$.

## 2.5 Ingredients of WGANTPP

To proceed with our point process based WGAN, we need the generator function $g_\theta : \Xi \to \Xi$, the discriminator function $f_w : \Xi \to \mathbb{R}$, and enforce Lipschitz constraint on $f_w$. Figure 4 in Appendix A illustrates the data flow for WGANTPP.

The generator transforms a given sequence to another sequence. Similar to [32, 33] we use Recurrent Neural Networks (RNN) to model the generator. For clarity, we use the vanilla RNN to illustrate the computational process as below. The LSTM is used in our experiments for its capacity to capture long-range dependency. If the input and output sequences are $\zeta = \{z_1, \ldots, z_n\}$ and $\rho = \{t_1, \ldots, t_n\}$ then the generator $g_\theta(\zeta) = \rho$ works according to

$$h_i = \phi_g^h(A_g^h z_i + B_g^h h_{i-1} + b_g^h), \qquad t_i = \phi_g^x(B_g^x h_i + b_g^x) \tag{7}$$

Here $h_i$ is the $k$-dimensional history embedding vector and $\phi_g^h$ and $\phi_g^x$ are the activation functions. The parameter set of the generator is $\theta = \left\{ \left(A_g^h\right)_{k \times 1}, \left(B_g^h\right)_{k \times k}, \left(b_g^h\right)_{k \times 1}, \left(B_g^x\right)_{1 \times k}, \left(b_g^x\right)_{1 \times 1} \right\}$. Similarly, we define the discriminator function who assigns a scalar value $f_w(\rho) = \sum_{i=1}^{n} a_i$ to the sequence $\rho = \{t_1, \ldots, t_n\}$ according to

$$h_i = \phi_d^h(A_d^h t_i + B_g^h h_{i-1} + b_g^h) \qquad a_i = \phi_d^a(B_d^a h_i + b_d^a) \tag{8}$$

where the parameter set is comprised of $w = \left\{ \left(A_d^h\right)_{k \times 1}, \left(B_d^h\right)_{k \times k}, \left(b_d^h\right)_{k \times 1}, \left(B_d^a\right)_{1 \times k}, \left(b_d^a\right)_{1 \times 1} \right\}$. Note that both generator and discriminator RNNs are causal networks. Each event is only influenced by the previous events. To enforce the Lipschitz constraints the original WGAN paper [18] adopts weight clipping. However, our initial experiments shows an inferior performance by using weight clipping. This is also reported by the same authors in their follow-up paper [27] to the original work. The poor performance of weight clipping for enforcing 1-Lipschitz can be seen theoretically as well: just consider a simple neural network with one input, one neuron, and one output: $f(x) = \sigma(wx + b)$ and the weight clipping $w < c$. Then,

$$|f'(x)| \leq 1 \iff |w\sigma'(wx + b)| \leq 1 \iff |w| \leq 1/|\sigma'(wx + b)| \tag{9}$$

It is clear that when $1/|\sigma'(wx + b)| < c$, which is quite likely to happen, the Lipschitz constraint is not necessarily satisfied. In our work, we use a novel approach for enforcing the Lipschitz constraints, avoiding the computation of the gradient which can be costly and difficult for point processes. We add the Lipschitz constraint as a regularization term to the empirical loss of RNN.

$$\min_{\theta} \max_{w \in \mathcal{W}, \|f_w\|_L \leq 1} \frac{1}{L} \sum_{l=1}^{L} f_w(\xi_l) - \sum_{l=1}^{L} f_w(g_\theta(\zeta_l)) - \nu \sum_{l,m=1}^{L} \left| \frac{|f_w(\xi_l) - f_w(g_\theta(\zeta_m))|}{|\xi_l - g_\theta(\zeta_m)|_\star} - 1 \right| \tag{10}$$

We can take each of the $\binom{2L}{2}$ pairs of real and generator sequences, and regularize based on them; however, we have seen that only a small portion of pairs $(O(L))$, randomly selected, is sufficient. The procedure of WGANTPP learning is given in Algorithm 1

**Remark** The significance of Lipschitz constraint and regularization (or more generally any capacity control) is more apparent when we consider the connection of W-distance and optimal transport problem [28]. Basically, minimizing the W-distance between the empirical distribution and the model distribution is equivalent to a semidiscrete optimal transport [28]. Without capacity control for the generator and discriminator, the optimal solution simply maps a partition of the sample space to the set of data points, in effect, *memorizing* the data points.

**Algorithm 1** WGANTPP for Temporal Point Process. The default values $\alpha = 1e - 4$, $\beta_1 = 0.5$, $\beta_2 = 0.9$, $m = 256$, $n_{\text{critic}} = 5$.

---

**Require:** : the regularization coefficient $\nu$ for direct Lipschitz constraint. the batch size, $m$. the number of iterations of the critic per generator iteration, $n_{\text{critic}}$. Adam hyper-parameters $\alpha, \beta_1, \beta_2$.
**Require:** : $w_0$, initial critic parameters. $\theta_0$, initial generator's parameters.
1: set prior $\lambda_z$ to the expectation of event rate for real data.
2: **while** $\theta$ has not converged **do**
3:      **for** $t = 0, ..., n_{\text{critic}}$ **do**
4:          Sample point process realizations $\{\xi^{(i)}\}_{i=1}^m \sim \mathbb{P}_r$ from real data.
5:          Sample $\{\zeta^{(i)}\}_{i=1}^m \sim \mathbb{P}_z$ from a Poisson process with rate $\lambda_z$.
6:          $L' \leftarrow \left[ \frac{1}{m} \sum_{i=1}^m f_w(g_\theta(\zeta^{(i)})) - \frac{1}{m} \sum_{i=1}^m f_w(\xi^{(i)}) \right] + \nu \sum_{i,j=1}^m |\frac{|f_w(\xi_i) - f_w(g_\theta(\zeta_j))|}{|\xi_i - g_\theta(\zeta_j)|_\star} - 1|$
7:          $w \leftarrow \text{Adam}(\nabla_w L', w, \alpha, \beta_1, \beta_2)$
8:      **end for**
9:      Sample $\{\zeta^{(i)}\}_{i=1}^m \sim \mathbb{P}_z$ from a Poisson process with rate $\lambda_z$.
10:      $\theta \leftarrow \text{Adam}(-\nabla_\theta \frac{1}{m} \sum_{i=1}^m f_w(g_\theta(\zeta^{(i)})), \theta, \alpha, \beta_1, \beta_2)$
11: **end while**

---

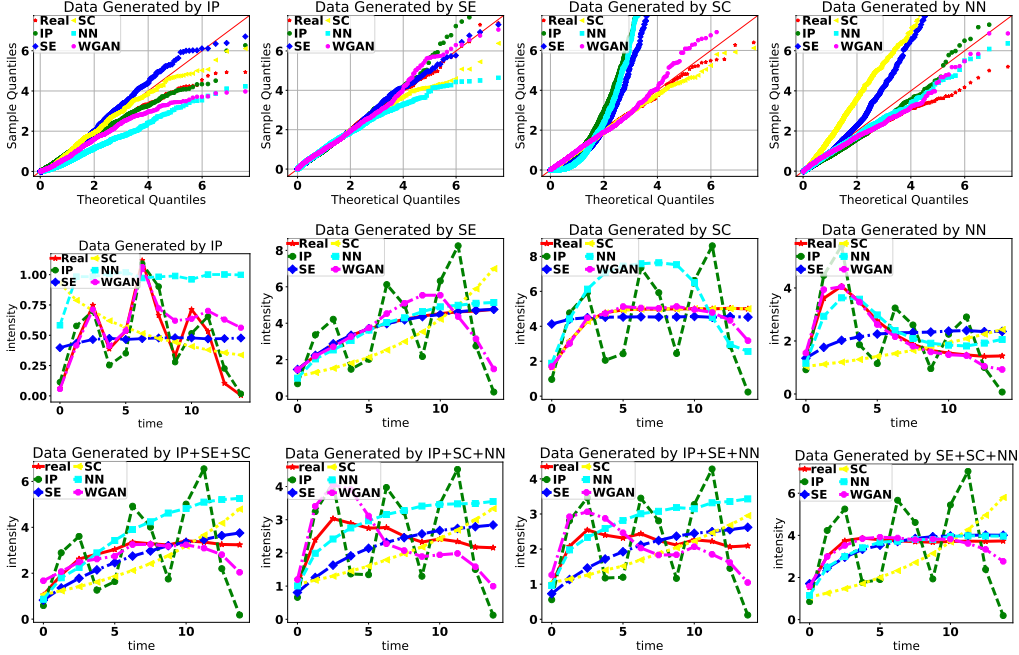

Figure 2: Performance of different methods on various synthetic data. Top row: QQ plot slope deviation; middle row: intensity deviation in basic conventional models; bottom row: intensity deviation in mixture of conventional processes.

# 3 Experiments

The current work aims at exploring the feasibility of modeling point process without prior knowledge of its underlying generating mechanism. To this end, most widely-used parametrized point processes, e.g., self-exciting and self-correcting, and inhomogeneous Poisson processes and one flexible neural network model, neural point process are compared. In this work we use the most general forms for simpler and clear exposition to the reader and propose the very first model in adversarial training of point processes in contrast to likelihood based models.

## 3.1 Datasets and Protocol

**Synthetic datasets.** We simulate 20,000 sequences over time $[0, T)$ where $T = 15$, for inhomogeneous process (IP), self-exciting (SE), and self-correcting process (SC), recurrent neural point process (NN). We also create another 4 ($= C_4^3$) datasets from the above 4 synthetic data by a uniform mixture

Table 1: Deviation of QQ plot slope and empirical intensity for ground-truth and learned model

| | Data | Estimator | | | | |
|---|---|---|---|---|---|---|
| | | MLE-IP | MLE-SE | MLE-SC | MLE-NN | WGAN |
| QQP. Dev. | IP | 0.035 (8.0e-4) | 0.284 (7.0e-5) | 0.159 (3.8e-5) | 0.216 (3.3e-2) | **0.033 (3.3e-3)** |
| | SE | 0.055 (6.5e-5) | **0.001 (1.3e-6)** | 0.086 (1.1e-6) | 0.104 (6.7e-3) | 0.051 (1.8e-3) |
| | SC | 3.510 (4.9e-5) | 2.778 (7.4e-5) | **0.002 (8.8e-6)** | 4.523 (2.6e-3) | 0.070 (6.4e-3) |
| | NN | 0.182 (1.6e-5) | 0.687 (5.0e-6) | 1.004 (2.5e-6) | 0.065 (1.2e-2) | **0.012 (4.7e-3)** |
| Int. Dev. | IP | **0.110 (1.9e-4)** | 0.241 (1.0e-4) | 0.289 (2.8e-5) | 0.511 (1.8e-1) | 0.136 (8.7e-3) |
| | SE | 1.950 (4.8e-4) | **0.019 (1.84e-5)** | 1.112 (3.1e-6) | 0.414 (1.6e-1) | 0.860 (6.2e-2) |
| | SC | 2.208 (7.0e-5) | 0.653 (1.2e-4) | **0.006 (9.9e-5)** | 1.384 (1.7e-1) | 0.302 (2.2e-3) |
| | NN | 1.044 (2.4e-4) | 0.889 (1.2e-5) | 1.101 (1.3e-4) | 0.341 (3.4e-1) | **0.144 (4.28e-2)** |
| Int. Dev. | IP+SE+SC | 1.505 (3.3e-4) | 0.410 (1.8e-5) | 0.823 (3.1e-6) | 0.929 (1.6e-1) | **0.305 (6.1e-2)** |
| | IP+SC+NN | 1.178 (7.0e-5) | 0.588 (1.3e-4) | 0.795 (9.9e-5) | 0.713 (1.7e-1) | **0.525 (2.2e-3)** |
| | IP+SE+NN | 1.052 (2.4e-4) | 0.453 (1.2e-4) | 0.583 (1.0e-4) | 0.678 (3.4e-1) | **0.419 (4.2e-2)** |
| | SE+SC+NN | 1.825 (2.8e-4) | 0.324 (1.1e-4) | 1.269 (1.1e-4) | 0.286 (3.6e-1) | **0.200 (3.8e-2)** |

from the triplets. The new datasets IP+SE+SC, IP+SE+NN, IP+SC+NN, SE+SC+NN are created to testify the mode dropping problem of learning a generative model. The parameter setting follows:

i) **Inhomogeneous process.** The intensity function is independent from history and given in Sec. 2.2, where $k = 3, \alpha = [3, 7, 11], c = [1, 1, 1], \sigma = [2, 3, 2]$.
ii) **Self-exciting process.** The past events increase the rate of future events. The conditional intensity function is given in Sec. 2.2 where $\mu = 1.0, \beta = 0.8$ and the decaying kernel $g(t - t_i) = e^{-(t-t_i)}$.
iii) **Self-correcting process.** The conditional intensity function is defined in Sec. 2.2. It increases with time and decreases by events occurrence. We set $\eta = 1.0, \gamma = 0.2$.
iv) **Recurrent Neural Network process.** The conditional intensity is given in Sec. 2.2, where the neural network's parameters are set randomly and fixed. We first feed random variable from [0,1] uniform distribution, and then iteratively sample events from the intensity and feed the output into the RNN to get the new intensity for the next step.

**Real datasets.** We collect sequences separately from four public available datasets, namely, healthcare MIMIC-III, public media MemeTracker, NYSE stock exchanges, and publications citations. The time scale for all real data are scaled to [0,15], and the details are as follows:

i) **MIMIC.** MIMIC-III (Medical Information Mart for Intensive Care III) is a large, publicly available dataset, which contains de-identified health-related data during 2001 to 2012 for more than 40,000 patients. We worked with patients who appear at least 3 times, which renders 2246 patients. Their visiting timestamps are collected as the sequences.
ii) **Meme.** MemeTracker tracks the *meme* diffusion over public media, which contains more than 172 million news articles or blog posts. The memes are sentences, such as ideas, proverbs, and the time is recorded when it spreads to certain websites. We randomly sample 22,000 cascades.
iii) **MAS.** Microsoft Academic Search provides access to its data, including publication venues, time, citations, etc. We collect citations records for 50,000 papers.
iv) **NYSE.** We use 0.7 million high-frequency transaction records from NYSE for a stock in one day. The transactions are evenly divided into 3,200 sequences with equal durations.

## 3.2 Experimental Setup

**Details.** We can feed the temporal sequences to generator and discriminator directly. In practice, all temporal sequences are transformed into time duration between two consecutive events, i.e., transforming the sequence $\xi = \{t_1, \ldots, t_n\}$ into $\{\tau_1, \ldots, \tau_{n-1}\}$, where $\tau_i = t_{i+1} - t_i$. This approach makes the model train easily and perform robustly. The transformed sequences are statistically identical to the original sequences, which can be used as the inputs of our neural network. To make sure we that the times are increasing we use elu + 1 activation function to produce positive inter arrival times for the generator and accumulate the intervals to get the sequence. The Adam optimization method with learning rate 1e-4, $\beta_1 = 0.5, \beta_2 = 0.9$, is applied. The code is available [2].

**Baselines.** We compare the proposed method of learning point processes (*i.e.*, minimizing sample distance) with maximum likelihood based methods for point process. To use MLE inference for point process, we have to specify its parametric model. The used parametric model are inhomogeneous Poisson process (mixture of Gaussian), self-exciting process, self-correcting process, and RNN. For

each data, we use all the above solvers to learn the model and generate new sequences, and then we compare the generated sequences with real ones.

**Evaluation metrics.** Although our model is an intensity-free approach we will evaluate the performance by metrics that are computed via intensity. For all models, we work with the empirical intensity instead. Note that our objective measures are in sharp contrast with the best practices in GANs in which the performance is usually evaluated subjectively, *e.g.*, by visual quality assessment. We evaluate the performance of different methods to learn the underlying processes via two measures: 1) The first one is the well-known QQ plot of sequences generated from learned model. The quantile-quantile (q-q) plot is the graphical representation of the quantiles of the first data set against the quantiles of the second data set. From the time change property [10] of point processes, if the sequences come from the point process $\lambda(t)$ , then the integral $\Lambda = \int_{t_i}^{t_{t+1}} \lambda(s)ds$ between consecutive events should be exponential distribution with parameter 1. Therefore, the QQ plot of $\Lambda$ against exponential distribution with rate 1 should fall approximately along a 45-degree reference line. The evaluation procedure is as follows: i) The ground-truth data is generated from a model, say IP; ii) All 5 methods are used to learn the unknown process using the ground-truth data; iii) The learned model is used to generate a sequence; iv) The sequence is used against the theoretical quantiles from the model to see if the sequence is really coming from the ground-truth generator or not; v) The deviation from slope 1 is visualized or reported as a performance measure. 2) The second metric is the deviation between empirical intensity from the learned model and the ground truth intensity. We can estimate empirical intensity $\lambda'(t) = E(N(t + \delta t) - N(t))/\delta t$ from sufficient number of realizations of point process through counting the average number of events during $[t, t + \delta t]$, where $N(t)$ is the count process for $\lambda(t)$. The $L_1$ distance between the ground-truth empirical intensity and the learned empirical intensity is reported as a performance measure.

### 3.3 Results and Discussion

**Synthetic data.** Figure 2 presents the learning ability of WGANTPP when the ground-truth data is generated via different types of point process. We first compare the QQ plots in the top row from the micro perspective view, where QQ plot describes the dependency between events. Red dots legend-ed with *Real* are the optimal QQ distribution, where the intensity function generates the sequences are known. We can observe that even though WGANTPP has no prior information about the ground-truth point process, it can estimate the model better except for the estimator that knows the parametric form of data. This is quite expected: When we are training a model and we know the parametric form of the generating model we can find it better. However, whenever the model is misspecified (*i.e.*, we don't know the parametric from *a priori*) WGANTPP outperforms other parametric forms and RNN approach. The middle row of figure 2 compares the empirical intensity. The *Real* line is the optimal empirical intensity estimated from the real data. The estimator can recover the empirical intensity well in the case that we know the parametric form where the data comes from. Otherwise, estimated intensity degrades considerably when the model is misspecified. We can observe our WGANTPP produces the empirical intensity better and performs robustly across different types of point process data. For MLE-IP, different number of kernels are tested and the empirical intensity results improves but QQ plot results degrade when the number of kernels increases, so only result of 3 kernels is shown mainly for clarity of presentation. The fact that the empirical intensity estimated from MLE-IP method are good and QQ plots are very bad indicates the inhomogeneous Poisson process can capture the average intensity (Macro dynamics) accurately but incapable of capturing the dependency between events (Micro dynamics). To testify that WGANTPP can cope with mode dropping, we generate mixtures of data from three different point processes and use this data to train different models. Models with specified form can handle limited types of data and fail to learn from diverse data sources. The last row of figure 2 shows the learned intensity from mixtures of data. WGANTPP produces better empirical intensity than alternatives, which fail to capture the heterogeneity in data. To verify the robustness of WGANTPP, we randomly initialize the generator parameters and run 10 rounds to get the mean and std of deviations for both empirical intensity and QQ plot from ground truth. For empirical intensity, we compute the integral of difference of learned intensity and ground-truth intensity. Table 1 reports the mean and std of deviations for intensity deviation. For each estimators, we obtain the slope from the regression line for its QQ plot. Table 1 reports the mean and std of deviations for slope of the QQ plot. Compared to the MLE-estimators, WGANTPP consistently outperforms even without prior knowledge about the parametric form of the true underlying generative point process. Note that for mixture models QQ-plot is not feasible.

**Real-world data.** We evaluate WGANTPP on a diverse real-world data process from health-care, public media, scientific activities and stock exchange. For those real world data, the underlying

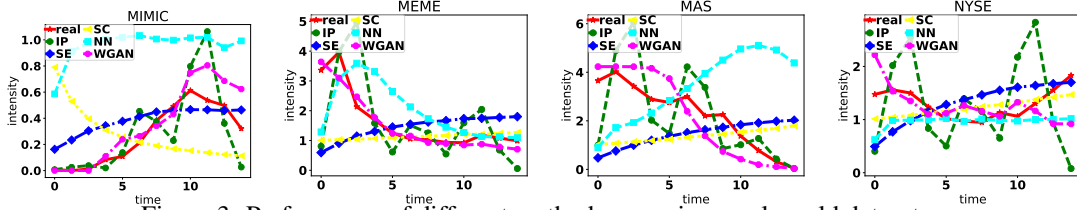

Figure 3: Performance of different methods on various real-world datasets.

Table 2: Deviation of empirical intensity for real-world data.

| Data | Estimator | | | | |
|------|---------|---------|---------|---------|---------|
|      | MLE-IP | MLE-SE | MLE-SC | MLE-NN | WGAN |
| MIMIC | 0.150 | 0.160 | 0.339 | 0.686 | **0.122** |
| Meme | 0.839 | 1.008 | 0.701 | 0.920 | **0.351** |
| MAS | 1.089 | 1.693 | 1.592 | 2.712 | **0.849** |
| NYSE | 0.799 | 0.426 | 0.361 | 0.347 | **0.303** |

generative process is unknown, previous works usually assume that they are certain types of point process from their domain knowledge. Figure 3 shows the intensity learned from different models, where *Real* is estimated from the real-world data itself. Table 2 reports the intensity deviation. When all models have no prior knowledge about the true generative process, WGANTPP recovers intensity better than all the other models across the data sets.

**Analysis.** We have observed that when the generating model is misspecified WGANTPP outperforms the other methods without leveraging the a priori knowledge of the parametric form. However, when the exact parametric form of data is known and when it is utilized to learn the parameters, MLE with this full knowledge performs better. However, this is generally a strong assumption. As we have observed from the real-world experiments WGANTPP is superior in terms of performance. Somewhat surprising is the observation that WGANTPP tends to outperform the MLE-NN approach which basically uses the same RNN architecture but trained using MLE. The superior performance of our approach compared to MLE-NN is another witness of the the benefits of using W-distance in finding a generator that fits the observed sequences well. Even though the expressive power of the estimators is the same for WGANTPP and MLE-NN, MLE-NN may suffer from mode dropping or get stuck in an inferior local minimum since maximizing likelihood is asymptotically equivalent to minimizing the Kullback-Leibler (KL) divergence between the data and model distribution. The inherent weakness of KL divergence [25] renders MLE-NN perform unstably, and the large variances of deviations empirically demonstrate this point.

## 4 Conclusion and Future Work

We have presented a novel approach for Wasserstein learning of deep generative point processes which requires no prior knowledge about the underlying true process and can estimate it accurately across a wide scope of theoretical and real-world processes. For the future work, we would like to explore the connection of the WGAN with the optimal transport problem. We will also explore other possible distance metrics over the realizations of point processes, and more sophisticated transforms of point processes, particularly those that are causal. Extending the current work to marked point processes and processes over structured spaces is another interesting venue for future work.

**Acknowledgements.** This project was supported in part by NSF (IIS-1639792, IIS-1218749, IIS-1717916, CMMI-1745382), NIH BIGDATA 1R01GM108341, NSF CAREER IIS-1350983, NSF CNS-1704701, ONR N00014-15-1-2340, NSFC 61602176, Intel ISTC, NVIDIA and Amazon AWS.

## Footnotes

[2]https://github.com/xiaoshuai09/Wasserstein-Learning-For-Point-Process

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
