[Supplementary Material · nips_2017_appendix.pdf]

# A   Data flow of Wassterstein learning for point process

Figure 4 illustrates the data flow for WGANTPP.

Figure 4: The input and output sequences are $\zeta = \{z_1, \ldots, z_n\}$ and $\rho = \{t_1, \ldots, t_n\}$ for generator $g_\theta(\zeta) = \rho$, where $\zeta \sim Poission(\lambda_z)$ process and $\lambda_z$ is a prior parameter estimated from real data. Discriminator computes the Wassterstein distance between the two distributions of sequences $\rho = \{t_1, t_2, \ldots\}$ and $\xi = \{\tau_1, \tau_2, \ldots\}$

.

# B   Proof that $\| \cdot \|_\star$ is a norm

It is obvious that $\| \cdot \|_\star$ is nonnegative and symmetric. If $\|\xi - \rho\|_\star = 0$, then $m = n$ and there is a assignment $\sigma$ such that $x_i = y_{\sigma(i)}$ for all $i = 1, \ldots, n$.

Now we prove that $\| \cdot \|_\star$ has triangle inequality. WLOG, assume that $\xi = \{x_1, \ldots, x_n\}$, $\rho = \{y_1, \ldots, y_k\}$ and $\zeta = \{z_1, \ldots, z_m\}$ where $n \leq k \leq m$. Define the permutation $\hat{\sigma}$ on $\{1, \ldots, k\}$ by

$$\hat{\sigma} := \arg\min_\sigma \sum_{i=1}^{n} \|x_i - y_{\sigma(i)}\| + \sum_{i=n+1}^{k} \|s - y_{\sigma(i)}\| \tag{11}$$

Then we know that

$$\|\xi - \rho\|_\star = \sum_{i=1}^{n} \|x_i - y_{\hat{\sigma}(i)}\| + \sum_{i=n+1}^{k} \|s - y_{\hat{\sigma}(i)}\| \tag{12}$$

Therefore, we have that

$$\|\xi - \zeta\|_\star = \min_\sigma \sum_{i=1}^n \|x_i - z_{\sigma(i)}\| + \sum_{i=n+1}^m \|s - z_{\sigma(i)}\|$$

$$\leq \min_\sigma \sum_{i=1}^n \left( \|x_i - y_{\hat\sigma(i)}\| + \|y_{\hat\sigma(i)} - z_{\sigma(i)}\| \right) + \sum_{i=n+1}^k \left( \|s - y_{\hat\sigma(i)}\| + \|y_{\hat\sigma(t)} - z_{\sigma(i)}\| \right)$$

$$+ \sum_{i=k+1}^m \|s - z_{\sigma(i)}\| \tag{13}$$

$$= \|\xi - \rho\|_\star + \min_\sigma \sum_{i=1}^k \|y_{\hat\sigma(i)} - z_{\sigma(i)}\| + \sum_{i=k+1}^m \|s - z_{\sigma(i)}\|$$

$$= \|\xi - \rho\|_\star + \min_\sigma \sum_{i=1}^k \|y_i - z_{\sigma(\hat\sigma^{-1}(i))}\| + \sum_{i=k+1}^m \|s - z_{\sigma(i)}\|$$

$$= \|\xi - \rho\|_\star + \|\rho - \zeta\|_\star$$

where the last equality is due to the fact that the minimization is taken over all permutations $\sigma$ of $\{1, \ldots, m\}$, and $\hat\sigma$ is a fixed permutation of $\{1, \ldots, k\}$ where $k \leq m$. This completes the proof.

## C  Proposed $\| \cdot \|_\star$ Distance on the Real Line

In this section, we prove that finding the distance between sequences $\xi$ and $\rho$,

$$\|\xi - \rho\|_\star = \min_\sigma \sum_{i=1}^n \|x_i - y_{\sigma(i)}\|_\circ + \sum_{i=n+1}^m \|s - y_{\sigma(i)}\|, \tag{14}$$

in the case of temporal point process in $[0, T)$, i.e., $\xi = \{t_1 < t_2 < \ldots < t_n\}$ and $\rho = \{\tau_1 < \tau_2 < \ldots < \tau_m\}$, reduces to

$$\|\xi - \rho\|_\star = \sum_{i=1}^n |t_i - \tau_i| + \sum_{i=n+1}^m (T - y_i), \tag{15}$$

Here, without loss of generality $n \leq m$ is assumed. The choice of $s = T$ is basically padding the shorter sequences with $T$. Given, the sequences have the same length now, we claim that the identity permutation i.e., $\sigma(i) = i$ is the minimizer in (14). We proceed by a proof by contradiction. Assume that the minimizer is NOT the identity permutation. Then, find the first $i$ such that $\sigma(i) \neq i$. Then, $\Sigma(i) = j$ where $j > i$. Therefore, there should be a $k > i$ such that $\sigma(k) = i$. Then, if you change the permutation according to $\sigma(i) = i$ and $\sigma(k) = j$ the cost will change by

$$\Delta = \underbrace{(|t_i - \tau_j| + |t_k - \tau_i|)}_{\text{for the old permutation}} - \underbrace{(|t_i - \tau_i| + |t_k - \tau_j|)}_{\text{for the new permutation}} \tag{16}$$

Given $i < j$ and $i < k$, it is easy to see that $\Delta > 0$. This means that we've found a better permutation which contradicts our assumption. Therefore, the optimal permutation will match the event points in an increasing order one by one.

## D  Equivalence of the $\| \cdot \|_\star$ Distance and Difference in Count Measures

The count measure of a temporal point process is a special case of the one defined for point processes in general space in Section 2.1. For a Borel subset $B \subset S = [0, T)$ we have $N(B) = \int_{t \in B} \xi(t) dt$. With a little abuse of notation we write $N(t) := N([0, t)) = \int_0^t \xi(t) dt$. Figure 1 is a good guidance through this paragraph. Starting from time 0 the first gap in count measure starts from $\min(t_1, \tau_1)$ and ends in $\max(t_1, \tau_1)$. Therefore, there is difference equal to $s_1 = \max(t_1, \tau_1) - \min(t_1, \tau_1) = |t_1 - \tau_1|$ in the count measure. Similarly, the second block of difference has volume of $s_2 = |t_2 - \tau_2|$, and so on. Finally, for $m > n$ the $(n + i)$-th block make a difference of $s_{n+i} = T - \tau_{n+i}$. Therefore, the area ($L_1$ distance) between the two sequences is a equal to $S = \sum_{i=1}^m s_i$. On the other hand by looking (15) we observe that $\|\xi - \rho\|_\star = \sum_{i=1}^m s_i$. Therefore, by choice of $s = T$ as an anchor point, the distance we have is exactly the area between the two count measures.