[Reviews · NeurIPS 2017]

Reviewer 1



The paper proposes using Wasserstein generative adversarial networks (WGAN) for point process intensity or hazard estimation. The paper demonstrates the usefulness of objective functions beyond MLE and proposes a computationally efficient distance measure. It uses a regularized dual formulation for optimization and an RNN for generator and discriminator functions. Details on the methods and data used in the real analyses are light and could be documented more clearly for reproducability. The results demonstrate that the WGANTPP performs reasonably in a variety of tasks using both simulated and real data. The comparison baselines appear unreasonably weak, e.g. the kernel mixture process uses 3 components and visibly has 3 humps in all results, and e.g. the self-exciting and self correcting processes use fixed baselines when there is a large literature on settings for semi-parametric hazard functions. Using more flexible and analogous comparison models, e.g. Goulding ICDM 2016, Weiss ECML 2013, Jing WSDM 2017, would make for better comparison. The distance defined would not deal with noise in the form of extraneous events due to the alignment of the ith element with the ith element of each sequence. Might a distance based on local alignment would likely be more robust than the one proposed? In what specific applications do you expect this distance to be more useful than log likelihood. Eq 3. perhaps the second summation should read \sum_{j=i+1}^m y_j. Section 3.1 inhomogeneous process: \alpha and c settings switched? Spelling e.g. demosntrate

Reviewer 2



This paper proposes to perform estimation of a point process using the Wasserstein-GAN approach. More precisely, given data that has been generated by a point process on the real line, the goal is to build a model of this point process. Instead of using maximum likelihood, the authors proposed to use WGAN. This requires to: - define a distance between 2 realizations of a point process - define a family of Lipschitz functions with respect to this distance - define a generative model which transforms "noise" into a point process The contribution of the paper is to propose a particular way of addressing these three points and thus demonstrate how to use WGAN in this setting. The resulting approach is compared on a variety of point processes (both synthetic and real) with maximum likelihood approaches and shown to compare favorably (especially when the underlying intensity model is unknown). I must admit that I am not very familiar with estimation of point processes and the corresponding applications and thus cannot judge the potential impact and relevance of the proposed method. However, I feel that the adaptation of WGAN (which is becoming increasingly popular in a variety of domains) to the estimation of point processes is not so obvious and the originality of the contribution comes from proposing a reasonable approach to do this adaptation, along with some insights regarding the implementation of the Lipschitz constraint which I find interesting. One aspect that could be further clarified is regarding the modeling of the generator function: from the definition in equation (7) there is no guarantee that the generated sequence t_i will be increasing. It is the case that the weights are constrained to be positive for example? or is it the case that the algorithm works even when the generated sequence is not increasing since the discriminator function would discriminate against such sequences and thus encourage the generator to produce increasing ones?

Reviewer 3



This paper presents a method for learning predictive (one-dimensional) point process models through modelling the count density using a W-GAN. Detailed comments: * Abstract: there are several approaches for doing full (or approximate) Bayesian inference on (Cox or renewal) point processes. E.g. see arxiv.org/pdf/1411.0254.pdf or www.gatsby.ucl.ac.uk%2F~vrao%2Fnips2011_raoteh.pdf. * To me it appears that the approach will only work for 1D point processes, as otherwise it is hard to represent them via the count density? If this is the case, it would be good to see this more explicitly stated. * This is the case for most inhomogenous point processes, but given only a single realisation of the point process, it would seem very hard to characterise a low variance estimate of distance from the generated count measure to the data? Perhaps this is the main reason that the W-GAN performs so well---much like the GP based intensity-explicit models, a heavy regularisation is applied to the generator/intensity-proxy. * I don't understand the origin of 'real' in the real world evaluation metrics e.g. Figure 3? How do you arrive at this ground truth? * A discussion of how easy/difficult these models are to train would have been interesting. Finally I am very interested to know how simple models compare to this: e.g. KDE with truncation, simple parametric Hawkes etc? My main concern with this work would be that these models are all horrendously over-complex for the signal-to-noise available, and that therefore while the W-GAN does outperform other NN/RNN based approaches, a more heavily regularised (read simpler) intensity based approach would empirically outperform in most cases.